# Advances in the Anti-Atherosclerotic Mechanisms of Epigallocatechin Gallate

**DOI:** 10.3390/nu16132074

**Published:** 2024-06-28

**Authors:** Yihui Liu, Yiling Long, Jun Fang, Gang Liu

**Affiliations:** College of Bioscience and Biotechnology, Hunan Agricultural University, Changsha 410128, China; liuyihui@stu.hunau.edu.cn (Y.L.); longyiling@stu.hunau.edu.cn (Y.L.); fangjun1973@hunau.edu.cn (J.F.)

**Keywords:** EGCG, cardiocerebrovascular diseases, atherosclerosis, mechanisms

## Abstract

Atherosclerosis (AS) is a common clinical sickness and the major pathological basis of ischemic cardiocerebrovascular diseases (CCVDs). The pathogenesis of AS involves a variety of risk factors, and there is a lack of effective preventive and curative drugs that can completely treat AS. In recent years, with the improvement of people’s living standards and changes in dietary habits, the morbidity and mortality rates of AS are on the rise, and the age of onset tends to be younger. The formation of AS is closely related to a variety of factors, and the main factors include lipid metabolism disorders, endothelial damage, inflammation, unstable plaques, etc. Epigallocatechin gallate (EGCG), as one of the main components of catechins, has a variety of pharmacological effects, and its role in the prevention of AS and the protection of cardiovascular and cerebral blood vessels has been highly valued. Recent epidemiological investigations and various in vivo and ex vivo experiments have shown that EGCG is capable of resisting atherosclerosis and reducing the morbidity and mortality of AS. In this paper, we reviewed the anti-AS effects of EGCG and its mechanisms in recent years, including the regulation of lipid metabolism, regulation of intestinal flora disorders, improvement of vascular endothelial cell functions, inhibition of inflammatory factors expression, regulation of inflammatory signaling pathways, inhibition of matrix metalloproteinase (MMP) expression, and inhibition of platelet aggregation, which are helpful for the prevention of cardiocerebrovascular diseases.

## 1. Introduction

Cardiocerebrovascular diseases are the manifestation of systemic vascular lesions in the heart and brain. Its etiology mainly includes atherosclerosis, hypertension, hyperlipidemia, diabetes mellitus, thrombocytosis, and other factors, and its morbidity, disability, and mortality rates are extremely high, causing serious damage to the life and health of patients, especially for middle-aged and elderly people [1]. Even with the most advanced and sophisticated medical treatments, more than 50% of cerebrovascular disease survivors are still unable to live adequately on their own. Each year, approximately 15 million people worldwide die from cardiocerebrovascular diseases, ranking them among the top causes of death [2]. Atherosclerosis (AS) is the underlying pathological manifestation of many cardiocerebrovascular diseases and is characterized by classical inflammatory degeneration, exudation, and proliferation [3]. Atherosclerotic plaques are not stable, and when ruptured, they block blood vessels in a short period, causing a predisposition to acute cardiovascular disease, which can be life-threatening [4]. Many people believe that to fight AS is to protect health and life, that the fundamental measure to combat the onset of cardiocerebrovascular diseases is to prevent AS from occurring, and that the prevention of AS is conducive to curbing the continued increase in mortality from cardiocerebrovascular diseases worldwide. Currently, the main therapeutic drugs for AS are statins (lipid-regulating drugs), ticagrelor (antiplatelet drugs), calcium antagonists (CCBs), etc. [5,6]. The causes of AS are complex, and it is difficult to prevent and control with drugs. Most of them have side effects, so early prevention and control are very important. EGCG is the main body of the components in catechins, which is known to have antihypertensive, lipid-lowering, hypoglycemic, weight-loss, anti-inflammatory, anti-oxidation, anti-aging, and other multiple functions. In recent years, epidemiological studies have shown that EGCG can prevent cardiocerebrovascular diseases, and it has a prominent role in anti-AS, which can reduce the risk of cardiocerebrovascular diseases [7]. In this paper, we will review the ameliorative effects of EGCG, a natural compound in tea, on atherosclerosis; explore its source, structure, and physiological functions; and focus on its ameliorative mechanisms on AS to provide a systematic reference for future research on EGCG in cardiocerebrovascular diseases.

## 2. Sources, Structure, and Physiological Functions of EGCG

### 2.1. Sources and Structure of EGCG

EGCG is found in abundance, mainly in tea, especially green tea, and it can contain up to 7380 mg per 100 g of dried tea according to the United States Department of Agriculture (USDA) [8]. In addition, small amounts of EGCG can also be detected in a wide range of plants, such as apples, cranberries, hazelnuts, carob beans, etc. The food sources and content of EGCG are shown in Table 1. EGCG is the most abundant component of catechins, accounting for about 50% to 70% of total catechins, and it is also the major component of the tea polyphenols that exert a variety of biological activities. Catechin is a phenolic active ingredient extracted from natural plants, such as tea. It belongs to the flavanols and is the most important tea polyphenol [9]. Catechins mainly consist of eight monomer types: epigallocatechin gallate (EGCG), epicatechin (EC), epicatechin gallate (ECG), epigallocatechin (EGC), catechin (C), gallocatechin (GC), catechin gallate (CG), and gallocatechin gallate (GCG) [10]. EGCG has the highest content in catechins. Due to its special structure, its efficacy is also more prominent, in the field of medicine, and functional food has broad prospects for application [11]. The molecular structure of EGCG is shown in Figure 1. EGCG can be formed by the esterification of EGC and GA (gallic acid), and it is more stable under weakly acidic conditions and unstable under neutral and alkaline conditions [12]. The EGCG molecule contains three aromatic rings, one pyran ring, and eight phenolic hydroxyls, and the phenolic hydroxyls in the structure can provide hydrogen atoms for the redox reaction. And the generated radicals contain the structure of catechols with a high degree of stability [13]. In addition, the *o*-dihydroxy catechol structure of the B-ring and the 2,3-double bond of dihydropyran in EGCG may be the site for accepting reactive oxygen species, and thus, it has a stronger antioxidant capacity than other catechins.

### 2.2. Physiological Functions of the EGCG

Many studies have shown that there are diverse pharmacological effects of EGCG (Figure 2), such as anti-inflammatory, antioxidant, antidiabetic, anti-obesity, and blood-pressure lowering, etc., among which the anti-inflammatory and antioxidant effects are considered to be the most important ones of EGCG. EGCG, as a natural antioxidant, has an antioxidant activity even higher than that of vitamin E. It also scavenges free radicals produced by the human body, protects cell membranes, and has a delayed-aging effect [14]. In terms of anti-inflammation, they can prevent vascular inflammation by decreasing the adhesion of leukocytes to endothelial cells and the production factors and adhesion molecules of cytochrome C mediated by nuclear transcription factors [15]. Some clinical trials have shown that EGCG can significantly reduce dental plaque and delay periodontal disease [16]. EGCG also has some antimicrobial effects, inhibiting pathogenic bacteria without affecting the reproduction of beneficial bacteria, and has inhibitory effects on both bacteria and fungi [17]. In addition, it can inhibit the bacteria that cause human skin diseases and is effective in the treatment of eczema. EGCG removes the odor of methanethiol and, therefore, reduces the smell of cigarette smoke in the mouths of smokers and has a deodorizing effect by resisting odor-producing bacteria in the human intestinal tract. In addition, EGCG can induce anti-angiogenic and anti-proliferative effects in cancer cells and has been helpful in the prevention of cancers, such as hepatocellular carcinoma, gastric cancer, lung cancer, and breast cancer. Its possible mechanisms of action include anti-proliferative, anti-migratory, pro-apoptotic, and anti-angiogenic effects [18].

## 3. Studies on the Mechanisms of EGCG against AS

### 3.1. Regulation of Lipid Metabolism

Dyslipidemia is one of the important predisposing factors of AS, and regulating blood lipid levels can prevent AS [19]. Low-density lipoprotein cholesterol (LDL-C), cholesterol (TC), and triglycerides (TG) are the main factors in the formation of atherosclerosis. Histomorphometric studies of early atherosclerotic injury have shown that the deposition of cholesterol, as well as other lipids within the vessel wall, is the initiating event in the overall process of atherosclerosis. And, these lipid deposits are derived primarily from blood lipoproteins, particularly LDL. Low-density lipoprotein (LDL) is the major lipoprotein in plasma for the transport of cholesterol and is converted from very low-density lipoprotein (VLDL), which consists mainly of apolipoprotein B-100 (apoB-100) and lipids. LDL mainly transports cholesterol from the liver to the outside of the body. If the LDL metabolism is disturbed, it will lead to an enhancement of plasma LDL, which in turn, affects cholesterol levels. Cholesterol accumulation in the walls of the tubes tends to lead to atherosclerosis. High levels of plasma LDL-C are an important factor leading to AS, and the reduction of LDL-C can degrade the early stages of atherosclerosis, which can reduce the incidence of cardiovascular events [20]. It has been shown that by feeding oolong tea and green tea leaves to rats, their body weight and their plasma triglycerides, cholesterol, and LDL were markedly reduced, with EGCG playing a major role [21]. Momose et al. [22] conducted a study on the ability of EGCG to lower LDL-C, made subjects ingest 107–856 mg/d EGCG for 4 to 14 weeks, and showed that EGCG intake led to a prominent reduction in LDL-C. In addition, it has been shown that catechins can inhibit cholesterol synthesis and promote cholesterol excretion, normalizing cholesterol levels and reducing lipid deposition [23]. Chen et al. [24] showed that, after 12 weeks of therapy with a high dose of EGCG (856.8 mg) administered daily to obese females, the weight of the treatment group was significantly reduced, total cholesterol decreased to 5.33%, and LDL plasma levels decreased. Ge Hu et al. [25] showed that EGCG inhibited cholesterol synthesis by down-regulating the activities of mevalonate kinase (MVK), mevalonate 5-pyrophosphate decarboxylase (MDD), and farnesyl pyrophosphate synthase (FPPS) in the mevalonate pathway. In addition, EGCG can reduce TG by directly inhibiting fatty acid synthase (FAS) activity or by down-regulating peroxisome proliferator-activated receptor γ (PPARγ) and FAS expression levels through the PI3K-AKT signaling [26].

LDL is a significant cause of AS plaque formation and secondary cardiovascular disease [27]. Oxidative stress, which causes reactive oxygen species (ROS) to accumulate in the body or cells and predispose to tissue damage, plays a key role in the occurrence of AS and is a key factor in the formation of AS, and it plays a critical role in the oxidation of LDL in the early stages of AS [28]. According to the oxidative stress theory of AS, ROSs can give rise to the oxidation of LDL to produce oxidized low-density lipoprotein (ox-LDL) [29]. Oxidative stress causes oxidative modification of a large number of LDLs, and ox-LDL plays a critical role in the development of AS. Studies have shown that ox-LDL is an independent risk factor for AS and is closely related to all stages of AS. It is capable of damaging vascular endothelial cells and promotes monocyte adhesion; VSMC migration; value-added, foam cell, and thrombus formation; and AS plaque lysis. The oxidative product ox-LDL is phagocytosed by monocyte macrophages, and lipid-carrying macrophages are converted to foam cells and accumulate, which can lead to the occurrence of AS [30]. Goto et al. [31] found that 25 μMol·L^−1^ of EGCG up-regulated the expression of the LDL receptor and lowered the apoB100 level to improve the metabolism and thus the metabolism of cholesterol. In vitro biochemical assays, catechins, especially EGCG, inhibited the oxidation of plasma LDL. Choi et al. [32] found that EGCG (25 μMol·L^−1^) inhibited ox-LDL-induced ROS production, thereby protecting human vascular endothelial cells. Li et al. [33] demonstrated that EGCG (10–100 μMol·L^−1^) decreased AngII-induced NADPH oxidase expression, lowered ROS generation, and repressed AngII-induced activation of NF-κB and activator protein-1 (AP-1). These results suggest that EGCG can reduce the incidence of AS by regulating blood lipids and inhibiting the oxidation of LDL, which is important for the prevention and therapy of AS (Figure 3).

### 3.2. Regulation of Intestinal Flora Disorders

Gut microbes can break down dietary fiber and metabolize it to produce diverse metabolites, which play a prominent role in regulating host digestion, absorption, and immune response. Changes in the composition and dysfunction of the gut flora have a meaningful impact on the occurrence and progression of AS [34]. The main probiotic bacteria in the intestinal flora include 23 genera of lactic acid bacteria, such as *Lactobacillus* spp., *Streptococcus* spp., and *Bifidobacterium* spp., as well as *Faecalibacterium prausnitzii* and *Akkermansia muciniphila*. Among them, *Akkermansia muciniphila* has important roles in immune modulation and anti-tumor, obesity suppression, and inflammation relief, and it is a new type of probiotic [35]. Intestinal flora can influence the development of atherosclerotic plaques by regulating cholesterol and lipid metabolism in the host. Numerous studies have shown that obesity and related metabolic diseases are associated with gut microbial disorders [36]. The gut flora of obese individuals can efficiently obtain energy from food and store it as fat, which tends to increase blood lipids, and long-term hyperlipidemia is very prone to the formation of AS. The digestive and catabolic function of gut microorganisms on high-fat foods tends to affect lipid metabolism, which further affects blood lipid levels and the occurrence of AS. EGCG plays a role in regulating the activity of gut microorganisms, which can reduce energy intake, lower body weight, regulate blood lipid levels, improve liver tissue damage and gut flora disorders, and have a good effect on weight loss [37]. EGCG has a good effect on regulating the cholesterol metabolite bile acids (BAs), which reduce the reabsorption of BAs, and this effect is mediated by intestinal microbes. It was shown that 0.32% EGCG increased the levels of cholesterol 7α-carboxylase (5.6-fold increase), HMG-CoA reductase, LDL receptor mRNA expression levels, decreased BAs reabsorption, and total fecal BAs excretion increased 1.5 fold, resulting in lower levels of intestinal BAs [38]. Sheng et al. [39] analyzed the impacts of EGCG on obesity in terms of bile acid signaling and adjusting intestinal flora. The results suggested that EGCG increased *Verrucomicrobiaceae*, promoted the proliferation of *Akkermansia muciniphila* in the gut, and raised the levels of farnesoid X receptor (FXR) and Takeda G protein receptor (TGR)-5 agonists and their regulatory signaling in the liver, affecting the metabolism of fat and glucose in vivo. It is evident that catechins decrease the levels of TG and TC in vivo by regulating the species and abundance of the associated flora to achieve hypolipidemic effects, thereby preventing the development of AS diseases. In addition, intestinal dysbiosis may lead to an inflammatory response that exacerbates the development of atherosclerotic plaques or leads to plaque rupture. It has been shown that GTPs (containing EGC, EGCG, etc.) lower the abundance of *Bacteroidetes* and *Fusobacteria* and add to the abundance of *Firmicutes*, decrease the expression of the inflammatory factors IL-6, TNF-α, and IL-1β, and restrain the induction level of the inflammatory signaling pathway toll-like receptor 4 (TLR4) [40]. Metabolites of gut flora can also exert positive (e.g., short-chain fatty acids (SCFAs)) or negative (e.g., trimethylamine oxide (TMAO)) effects on the development of atherosclerosis. *Faecalibacterium prausnitzii* of Firmicutes can generate the anti-inflammatory metabolites SCFAs, which are one of the key bacteria for the therapy of inflammation. Oral EGCG can also realize the same impact by enriching SCFA-producing bacteria [41].

### 3.3. Improvement of Vascular Endothelial Cells Functions

Vascular endothelial cells (ECs) are momentous for healthy cardiovascular homeostasis, and they also have a significant role in the pathological mechanisms of AS [42]. Apoptosis of vascular ECs is an early event in the occurrence of AS and promotes AS lesion formation, plaque erosion, and acute coronary syndromes [43]. Multiple risk factors for endothelial cell injury and pre-AS formation can induce endothelial cell apoptosis. EGCG can delay the progression of AS by restraining EC apoptosis, which is connected with the inhibition of activation and aberrant expression of the caspase family, which are key components in the apoptotic process, and whose activation and aberrant expression can lead to apoptosis. The B-cell lymphoma-2 (BCL-2) family of genes is associated with apoptosis, and they can inhibit it by regulating the release of cytochrome C and other proteins in mitochondria apoptosis. But, its BCL2-associated X (BAX) gene component can promote apoptosis [44]. Ox-LDL can lead to the overexpression of BAX genes, reduce the expression of BCL-2 and BCL-XL, and promote apoptosis [45]. EGCG can effectively restrain ox-LDL-mediated apoptosis in vascular ECs. In the H_2_O_2_-induced apoptosis model of rat smooth muscle cells, EGCG (10–150 μMol·L^−1^) significantly decreased the expression of the BAX gene, raised the expression of the BCL-2 gene, and declined the H_2_O_2_-induced expression of caspase-3, caspase-8, and caspase-9 [46]. Meng [47] et al. investigated the effect of EGCG on the hydrogen peroxide (H_2_O_2_)-induced endothelial injury model. Human vascular endothelial cells (HUVECs) were pretreated with different concentrations of EGCG and then exposed to H_2_O_2_. Cell viability was determined by an MTS assay. Apoptosis was detected by TUNEL staining, and apoptosis-related proteins were detected by Western blot. The results showed that EGCG pretreatment markedly increased the survival of HUVEC in H_2_O_2_-induced cell death. After exposure to H_2_O_2_, EGCG up-regulated the levels of Atg5, Atg7, LC3 II/I, and Atg5-Atg12 complexes in HUVEC, while down-regulating apoptosis-related proteins to achieve the efficacy of inhibiting apoptosis.

Vascular endothelial dysfunction strongly induces cardiovascular disease, which is common in AS. Nitric oxide (NO), as a momentous endothelium-derived diastolic factor, is associated with many key factors in the AS process, such as LDL oxidation, endothelial leukocyte adhesion, and VSMC proliferation, and plays a role in cardiocerebrovascular protection and anti-AS function. However, in pathological states, endothelial-type nitric oxide synthase (eNOS) disturbances lead to aberrant NO production, which impairs endothelial functions and triggers AS [48]. Regulating the balance of NO and endothelin helps to maintain normal vascular function and reduce the incidence of AS. EGCG can lead to an increase in endothelial-type nitric oxide synthase (eNOS) activity, an increase in NO production, and a diminution in blood pressure. Increasing the release of NO delays the hemodynamic changes in the endothelium in vivo, thus avoiding the strong stress response induced by the endothelium [49]. It has been shown that EGCG increases the bioavailability of normal NO by decreasing the levels of the endogenous NO inhibitor, asymmetric dimethylarginine. In addition, EGCG inhibits enhanced oxidative stress through the Nrf2/HO-1 pathway. These effects suggest that it may block the production of ROS, inhibit inflammation, and diminish EC apoptosis in the early stages of AS [50]. EGCG protects vascular endothelial cells from oxidative stress-induced injury by targeting the autophagy-dependent PI3K-AKT-mTOR pathway [47]. Xuan et al. [51] found that EGCG (10 mg·L^−1^) could protect ECs by activating the PI3K/Akt signaling pathway, increasing the NO level of vascular ECs and decreasing the level of caspase-3 in the cells. Other studies have shown that EGCG improves endothelial function by increasing eNOS production in vascular ECs through the PI3K/Akt signaling pathway, increasing the amount of NO in the cells and improving insulin sensitivity. Liu et al. [52] found that homocysteine (Hcy) inhibited the production of eNOS in HUVECs, whereas EGCG (3–30 μMol·L^−1^) could significantly ameliorate this phenomenon and inhibit intracellular caspase-3 and caspase-9 expression. In conclusion, EGCG can improve the function of vascular endothelial cells and further inhibit the occurrence of atherosclerosis by inhibiting EC apoptosis and up-regulating the level of NO in ECs.

### 3.4. Inhibition of Inflammatory Factors Expression

Chronic inflammation is one of the most significant causes of atherosclerosis [53]. A large number of experimental studies and clinical observations have confirmed that damaged endothelial cells secrete a variety of inflammatory factors as well as growth factors, which induce monocyte macrophages to adhere to ECs through the immunoglobulin superfamily and integrin family and then enter under the ECs, which is the causative factor of AS. Many lipids can act as signaling molecules that bind to receptors on ECs and activate the gene expression of many pro-inflammatory cytokines, such as monocyte chemotactic protein-l (MCP-1), vascular cell adhesion molecule-1 (VCAM-1), intercellular adhesion molecule-1 (ICAM-1), and endothelial cell leukocyte adhesion molecule (E-selectin), which play a significant role in the inflammatory process [54]. Activated macrophages secrete pro-inflammatory cytokines, such as TNF-α, IL-1β, IL-6, IL-4, and CRP, which are involved in the formation of inflammation and further contribute to the development of AS. One of the main mechanisms of anti-inflammation is the down-regulation of pro-inflammatory factors, which reduces the degree of inflammation or inhibits the development of inflammation. Studies have shown that inflammatory factors, such as IL-4 and IL-6, can be significantly suppressed by the oral administration of catechins in a rat model of atherosclerosis [55]. MCP-1 is a potent chemotactic protein for monocytes, basophils, and memory T cells and is also a potent chemotactic protein for endothelial leukocytes. Cells and elevated serum levels of MCP-1 are some of the inflammatory hallmarks leading to coronary heart illness. Wang et al. [56] showed that EGCG (10, 25, and 50 μMol·L^−1^) inhibited TNF-α-induced MCP-1 production in HUVEC cells, and the inhibitory effect showed a dose-dependent relationship. CRP, as an inflammatory cytokine, is directly involved in atherosclerotic plaque formation. Ramesh et al. [57] found that feeding rats an AS-causing diet and EGCG (100 mg/kg) for 45 days significantly inhibited CRP expression and decreased the erythrocyte sedimentation rate, total leukocyte count, and other inflammatory hematological parameters, suggesting that EGCG can inhibit AS formation by suppressing inflammation. VCAM-1 and ICAM-1 play a critical role in the adhesion of leukocytes to the surface of vascular endothelial cells and their entry beneath the endothelium, as well as in the proliferation of smooth muscle cells. Chae et al. [58] found that a 10–50 μMol·L^−1^ EGCG treatment significantly inhibited Angiotensin II (Ang II)-induced VCAM-1 and ICAM-1-related mRNA synthesis in HUVECs, as well as reduced VCAM-1 and ICAM-1 molecules in HUVECs membranes. Ludwig et al. [59] found that EGCG (10~100 μMol·L^−1^) dose-dependently inhibited VCAM-1 expression in HUVEC cells induced by IL-1β and inhibited TNF-α-induced adhesion of THP-1 monocytes to vascular ECs.

### 3.5. Regulation of Inflammatory Signaling Pathways

EGCG can affect many intracellular inflammatory signaling pathways associated with AS (Figure 4). Current research has shown that EGCG mainly acts on the nuclear factor kappa-light-chain-enhancer of activated B cells (NF-κB) signaling pathway, mitogen-activated protein kinase (MAPK) signaling pathway, notch signaling pathway, and so on. Research has shown that the NF-κB signaling pathway is a central link and common pathway that regulates the transcription of a variety of inflammatory factors, and NF-κB plays a key role in inflammatory diseases, as well as in inflammation-related disorders such as AS [60]. EGCG can partially inhibit the NF-κB signaling pathway, markedly down-regulate the expression of inflammatory factors, including adhesion molecules, cytokines, and MMPs, and exert an anti-AS effect. Wang et al. [56] found that EGCG blocked the 67 kDa laminin receptor (67LR)-mediated NF-κB signaling pathway in HUVEC cells, thereby inhibiting the production of the inflammatory factor MCP-1 in HUVEC cells. The MAPK signaling pathway plays a significant role in the regulation of cell proliferation, differentiation, transformation, and apoptosis, and the three kinases in the pathway play key roles in endothelial damage and protection These include extracellular signal-regulated kinase 1/2 (ERK1/2), c-Jun N-terminal protein kinase (JNK), and p38 mitogen-activated protein kinases (p38 MAPK). ERK can be activated by the phosphorylation of the upstream molecule MEK to further regulate cell differentiation and proliferation. P38 MAPK and JNK kinase activation, on the other hand, originate from the stress response and mediate cellular stress and apoptosis, with a mutual activation network formed between p38 MAPK and NF-κB regulating the gene expression of various inflammatory mediators and promoting the development of AS. EGCG interacts with 67LR and inhibits the activation of the ERK1/2, p38, and JNK signaling pathways. Yang et al. [61] found that, in an AngII-induced HUVEC cell model, EGCG (5–25 μMol·L^−1^) restrained the Ang II-induced activation of p38 MAPK and JNK1/2 kinases and reduced AngII-induced endothelial dysfunction. Chae et al. [58] found that EGCG (10–50 μMol·L^−1^) inhibited the phosphorylation of p38 MAPK and ERK in the MAPK pathway by inhibiting the p38 MAPK signaling pathway against HUVEC inflammation and adhesion.

Toll-like receptor 4 (TLR4) is a member of the TLR family, which participates in the inflammatory response of the body and is associated with the expression of inflammatory factors. TLR4 is expressed throughout the AS process, mainly in diseased macrophages and endothelial cells, and is involved in all stages of AS [62]. Studies have shown that, when TLR receptors are activated, they can express and secrete a variety of pro-inflammatory cytokines, such as TNF-α, IL-6, etc., and activate NF-κB through signaling. Therefore, inhibiting the TLR pathway can also prevent the activation of the NF-κB pathway and play an anti-inflammatory role. Hong et al. [63] found that 1 μMol·L^−1^ EGCG decreased the expression of TLR4 in macrophages. The downstream signaling pathway activation induced by EGCG on LPS was significantly attenuated by 67LR inhibitor or RNAi-mediated 67LR silencing. EGCG induced a significant up-regulation of Toll interaction protein (Tollip), a negative regulator of TLR signaling, which suggests an anti-inflammatory effect of EGCG. The notch signaling pathway plays an important role in the inflammatory response in AS, and it has a cross action with the PI3K/Akt and NF-κB signaling pathways, which can mediate cellular communication and regulate the immune response in AS [64]. Studies have shown that the number of notch ligands and receptors is increased in damaged myocardium and blood vessels, and EGCG can directly bind to notch receptors and inhibit the activation of the notch signaling pathway [65]. Yin et al. [66] demonstrated that in high-fat diet (HFD)-induced atherosclerosis, EGCG restrained the high-fat diet-induced inflammatory response in apolipoprotein E (ApoE) knockout (ApoEKO) mice via the jagged-1/notch pathway. Xie et al. [67] investigated the effect of EGCG on uric acid-induced HUVEC inflammation and found that EGCG (20 μMol·L^−1^) effectively inhibited the intracellular notch signaling pathway; markedly inhibited the secretion of IL-6, MCP-1, and TNF-α; and reduced ROS generation, which further inhibited inflammation. Huang et al. [68] evidenced that EGCG could directly bind to mouse notch-1; found that EGCG inhibited macrophage aggregation, inflammatory responses, and notch signaling in mice; and found that EGCG in mouse macrophages inhibited LPS-induced inflammatory responses, including over-activated notch signaling.

### 3.6. Inhibition of Matrix Metalloproteinase (MMP) Expression

Vascular smooth muscle cells (VSMCs) are the main cell type in all stages of AS plaques, and the pathological proliferation of VSMCs is a significant factor in the occurrence of AS. Thus, smooth muscle cell injury plays a critical role in AS [69]. Numerous studies have shown that some growth factors, such as PDGF, VEGF, and Ang II, can promote the growth of VSMCs and play a significant role in the course of AS. Matrix metalloproteinase (MMP), a superfamily of proteases with extracellular matrix-degrading activity, can increase the instability of AS plaques and further accelerate the development of AS [70]. During AS, inflammatory cell infiltration, VSMC migration, and proliferation are often accompanied by elevated matrix metalloproteinase activity. MMP-2 and MMP-9 are two major members of the MMP family [71]. Matrix metalloproteinase-2 (MMP-2), also known as gelatinase A, is capable of degrading a variety of collagen, gelatin, and basement membrane components, and plays a key role in the migration and proliferation of VSMCs. Matrix metalloproteinase-9 (MMP-9), also known as gelatinase B, is one of the major enzymes that degrades the extracellular matrix. MMP-9 is expressed mainly in smooth muscle cells, vascular endothelial cells, and macrophages at the base of plaques, AS plaques, and atherosclerotic injuries. In addition, extracellular MMP-inducible factor (EMMPRIN) is a transmembrane glycoprotein belonging to the immunoglobulin superfamily, which is widely expressed in a variety of cells in the human body [72]. EMMPRIN is closely related to the development of unstable plaques and acute coronary syndromes, and the expression of EMMPRIN in macrophages and VSMCs in plaques of patients with acute coronary syndromes is significantly increased [73]. Studies have shown that EGCG can prevent the migration of VSMCs by repressing the expression of MMP, and EGCG therapy markedly reduces the expression levels of MMP-2, MMP-9, and EMMPRIN [74]. Li et al. [75] investigated the effect of EGCG at a concentration of 1µM on lipopolysaccharide (LPS)-induced MMP-9 and MCP-1 expression in macrophages and its potential mechanism of action and showed that EGCG (1 µM) inhibited the TLR4/MAPK/NF-κB signaling pathway, reducing plaque instability and inhibiting the expression of MMP-9 and MCP-1, further stabilizing atherosclerotic plaques. Kim et al. [76] found that epigallocatechin gallate resulted in p21/WAF1-mediated cell cycle G(1) phase arrest and inhibited TNF-α-induced matrix metalloproteinase-9 expression in vascular smooth muscle cells. Bolduc et al. [77] used drinking-water-containing catechins (30 mg·kg^−1^-d^−1^) to feed spontaneous atherosclerotic (ATX) LDLr-/-: hApoB+/+ mice for 3 months and found that the catechins inhibited the effects of ROS in ATX mice. ROS induced MMP-9 activation and VSMC proliferation in ATX mice. Cheng et al. found that EGCG (1–10 μMol·L^−1^) inhibited the activation of MMP-2 in human aortic vascular smooth muscle cells and up-regulated the expression of the tissue inhibitory factor of MMP-2, TIMP-2 protein [78].

### 3.7. Inhibition of Platelet Aggregation

Platelets are the smallest blood cells in the blood and are the main component of the body for normal hemostatic function, and they are also essential for the activation of the coagulation system [79]. However, excessive platelet activation underlies the pathogenesis of many cardiocerebrovascular diseases and is a trigger for inflammation and atherosclerosis, so inhibition of platelet activation is very critical in the prevention and therapy of AS [80]. Platelet activation includes three processes: platelet adhesion, aggregation, and release. Among them, platelet aggregation is the key to hemostasis and thrombosis, and nowadays, many drugs that inhibit platelet aggregation have been used to treat thrombotic diseases [81]. Platelet glycoprotein GP IIb/IIIa binds to fibrinogen in response to certain stimulatory factors (e.g., thrombin, epinephrine, and thromboxane A2) and promotes platelet aggregation, leading to thrombosis and fibrin deposition. Antiplatelet therapy can reduce local thrombosis and inhibit vascular inflammation, thereby stabilizing vulnerable AS plaques. Studies have demonstrated that EGCG has antiplatelet aggregation and antithrombotic functions [82]. Lill et al. [83] investigated the effects of platelets with six catechins, including C, EC, EGC, CG, ECG, and EGCG, and found that only EGCG inhibited platelet aggregation induced by thrombin in vitro, while the other five catechins failed to restrain platelet aggregation. Ok et al. [84] found that ingestion of EGCG significantly restrained the increase of platelet thromboxane A2 in rats, which led to the suppression of collagen-induced platelet aggregation and modulation of inflammation. Most cells in the body, including platelets, metabolize arachidonic acid (AA), which in vivo produces the endoperoxides of prostaglandins PGG2 and PGH2 in response to platelet COX-1, the latter of which is capable of forming thromboxane A2 in response to thromboxane synthase. Studies have shown that EGCG inhibits cyclooxygenase-1 (COX-1) activity more strongly than aspirin, and in platelet aggregation experiments in collagen-induced rats, EGCG inhibited the production of thromboxane A2 by decreasing the activity of COX-1 and exhibited antiplatelet agglutination [85]. EGCG has been reported to inhibit platelet activity by several mechanisms, including the inhibition of collagen-mediated phospholipase (PL) Cgamma2, blockade of protein tyrosine phosphorylation, and enhancement of Ca^2(+)^-ATPase activity, thereby reducing platelet aggregation and attenuating thrombosis [86]. In addition, Joo et al. [87] activated and induced platelet aggregation by different platelet agonists, which increased platelet shear stress and caused platelet adhesion and simultaneously researched the antiplatelet effects of EGCG and antiplatelet drugs in vitro, and found that EGCG dose-dependently reduced platelet shear stress and inhibited platelet aggregation and adhesion.

## 4. Summary of EGCG Anti-AS Mechanisms and Shortcomings

EGCG can protect cardiocerebrovascular function by resisting atherosclerosis in several ways (Figure 5). Figure 5 clearly summarizes the mechanisms of action of EGCG against AS. These mechanisms include the reduction of blood lipid levels and LDL oxidation, regulation of intestinal flora disorders, improvement of vascular endothelial cells functions, reduction of inflammatory factors expression, regulation of signaling pathways, and suppression of MMP expression and platelet aggregation, suggesting that EGCG can resist atherosclerosis through multiple pathways, which is of great significance for the protection against cardiocerebrovascular diseases. Nowadays, many in vitro studies have shown that EGCG plays a significant role in cardiocerebrovascular diseases, especially in anti-atherosclerosis. However, as far as the current data are concerned, there are still a lot of issues that need to be addressed. For example, few products have been developed with EGCG as an active ingredient or as a food functional factor, as well as whether the ameliorative function of EGCG in AS is still controversial in populations of different ages and genders. In terms of toxicity, EGCG is safe as a natural product, having low toxicity, low side effects, and good water solubility [88]. However, the application of EGCG in the therapy of diseases has many problems. First, the chemical structure of EGCG is unstable and easily affected by conditions such as high temperature, pH, light, and metal ions [89]. Second, the bioavailability of EGCG in the body is low, making it difficult to maximize the efficacy of the drug, and the bioavailability of EGCG varies between species [90]. Third, the absorption rate of EGCG is easily affected by other foods or drugs [91]. Fourth, EGCG has a relatively short duration of action in the body; therefore, it cannot perform its function well [92]. However, in general, the study of the mechanism of EGCG anti-AS has positive research significance and value. Studies have shown that nanocarriers have proven to be excellent materials for encapsulating phenolic compounds and enhancing their bioavailability, including lipid nanoparticles, protein nanoparticles, micelles, emulsions, and metal nanoparticles [93]. By loading phenolic compounds into nanoparticles, not only can their bioavailability be improved, but also targeted release and protection of the active substance can be achieved [94]. Despite the fact that nanoparticles are almost a perfect carrier, there is still a need to consider and minimize their toxicity and side effects. In order to better use catechins for the prevention of cardiocerebrovascular diseases, it is necessary to further study the mechanisms of EGCG anti-AS and to increase the research on chemical modification of EGCG or changing its administration mode, so as to make it function optimally.

## 5. Conclusions

AS is a lipid-associated chronic inflammatory immune illness in which a large number of inflammatory immune cells accumulate in atheromatous plaques, thereby causing vasculopathy, microcirculatory disorders, and changes in blood rheology, which ultimately leading to cardiovascular and cerebral vascular lesions, severely affecting the health and quality of life of human beings. EGCG, as an active ingredient in natural plants, is of great research value. A growing number of reports suggest that EGCG can prevent and treat the development of AS, mainly due to its potential antioxidant and anti-inflammatory effects. However, caution needs to be exercised in its clinical use, and much in-depth research is required to fully understand its molecular role in various cells. In addition, the synergistic effect of EGCG with other drugs can also exert a certain anti-disease function, and in future research, it should also be further investigated whether its combined effect with other drugs can reduce the side effects and so on. At present, EGCG has been generally used in food, daily necessities, and other fields, and there are also reports on its antibacterial, deodorant, antiradiation, and antimutagenic properties, but its mechanism of action needs to be further studied. With the in-depth study of the pharmacological mechanisms of EGCG in vivo and in vitro, more and more functions of EGCG have been explored, and the application field can be further expanded, which has considerable social benefits for the development and therapy of a variety of illnesses.

## Figures and Tables

**Figure 1 nutrients-16-02074-f001:**
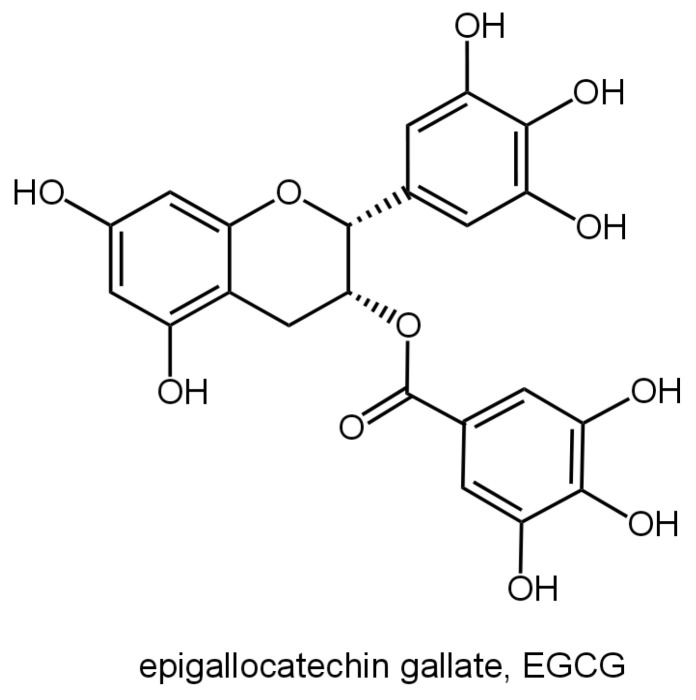
Molecular structure of EGCG.

**Figure 2 nutrients-16-02074-f002:**
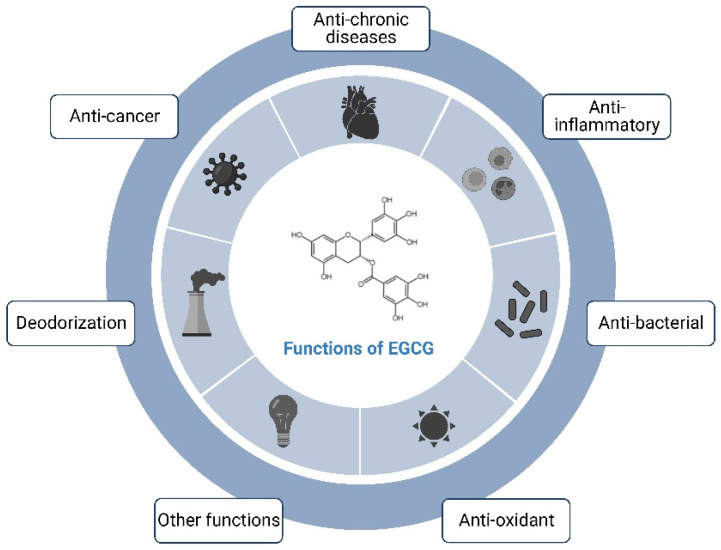
Schematic diagram of the functions of EGCG.

**Figure 3 nutrients-16-02074-f003:**
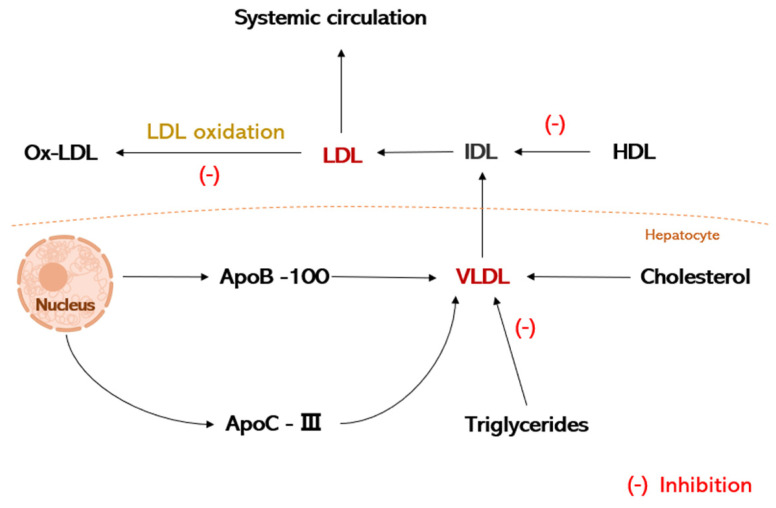
Schematic diagram of the mechanism of action of EGCG in lipid regulation.

**Figure 4 nutrients-16-02074-f004:**
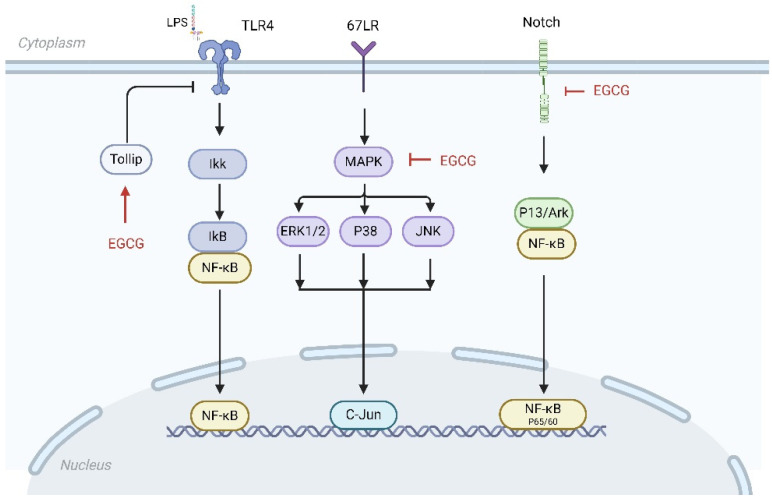
Schematic diagram of the regulatory effects of EGCG on MAPK, NF-κB, TLR4, and notch signaling pathways.

**Figure 5 nutrients-16-02074-f005:**
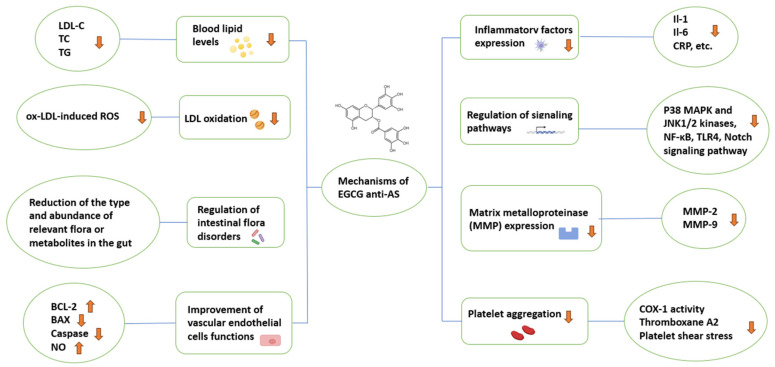
Schematic diagram of the anti-AS mechanisms of EGCG.

**Table 1 nutrients-16-02074-t001:** Table of food sources and content of EGCG.

Food Sources	Types	EGCG Content (mg/100 g)
Tea	Green	7380 (dry leaves)
		64.15 (brew)
	White	4245 (dry leaves)
		46.00 (brew)
	Oolong	34.48 (brew)
	Black	9.36 (brew)
Nuts	Hazelnuts	1.06
	Pistachio nuts	0.40
	Pecans	2.30
	Carob flour	109.46
Fruit	Apples	1.93
	Blackberries	0.68
	Cranberries	0.97
	Kiwifruit	0.09
	Peaches	0.30
	Pears	0.17
	Plums	0.40
	Strawberries	0.11
Vegetables	Onions	0.08

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
