# Peer review of "Advances in the Anti-Atherosclerotic Mechanisms of Epigallocatechin Gallate"

_nutrients, 2024, doi:10.3390/nu16132074_

Round 1
Reviewer 1 Report
Comments and Suggestions for Authors
This manuscript by Yihui Liu and colleagues reviewed the anti-atherosclerotic mechanisms of EGCG. The authors described EGCG structural, functional, and mechanistic insights. The authors also explained EGCG as a potential anti-atherosclerosis agent. This review in general is interesting. I have a minor concern: The acronym for Atherosclerosis is confusing, therefore please remove. I am not familiar with the word Cardiocerebraovascular in the field. Can you simply term as "Cardiovascular diseases and cerebrovascular diseases"
This review is nicely written, the graphical illustrations are useful.
Comments on the Quality of English LanguageEnglish editing is required.
Author Response
Dear reviewer,
Thank you for your careful and kind feedback on our manuscript. We have revised the manuscript based on your comments and have carefully proofread the manuscript to minimize typographical, grammatical, and referencing errors. The changes made in response to the reviewers' comments are detailed below.
Comment from Reviewer:
This manuscript by Yihui Liu and colleagues reviewed the anti-atherosclerotic mechanisms of EGCG. The authors described EGCG structural, functional, and mechanistic insights. The authors also explained EGCG as a potential anti-atherosclerosis agent. This review in general is interesting. I have a minor concern: The acronym for Atherosclerosis is confusing, therefore please remove. I am not familiar with the word Cardiocerebraovascular in the field. Can you simply term as "Cardiovascular diseases and cerebrovascular diseases"
This review is nicely written, the graphical illustrations are useful.
Response: Firstly, we would like to thank you for your endorsement of this review and for your suggestions We have removed the abbreviated word EGCG from the title. In addition, with regard to your suggestion of the term cardiocerebrovascular diseases, it is a collective term for cardiovascular and cerebrovascular diseases, which includes both, and is abbreviated as CCVDs. This word has been pointed out in a number of articles, and I have listed below some of the literature that has a source for the term cardiocerebrovascular diseases:
1.Hou Z, Lin Y, Yang X, Chen J, Li G. Therapeutics of Extracellular Vesicles in Cardiocerebrovascular and Metabolic Diseases. Adv Exp Med Biol. 2023;1418:187-205. doi: 10.1007/978-981-99-1443-2_13. PMID: 37603281.
2.Yang Y, Li Y, Wang J, Sun K, Tao W, Wang Z, Xiao W, Pan Y, Zhang S, Wang Y. Systematic Investigation of Ginkgo Biloba Leaves for Treating Cardio-cerebrovascular Diseases in an Animal Model. ACS Chem Biol. 2017 May 19;12(5):1363-1372. doi: 10.1021/acschembio.6b00762. Epub 2017 Apr 3. PMID: 28333443.
3.Liu B, Wang L, Jiang W, Xiong Y, Pang L, Zhong Y, Zhang C, Ou W, Tian C, Chen X, Liu SM. Myocyte enhancer factor 2A delays vascular endothelial cell senescence by activating the PI3K/p-Akt/SIRT1 pathway. Aging (Albany NY). 2019 Jun 10;11(11):3768-3784. doi: 10.18632/aging.102015. PMID: 31182679; PMCID: PMC6594820.
Reviewer 2 Report
Comments and Suggestions for Authors
1. The authors review the role of EGCG in Atherosclerosis. In most parts of their paper they only present results from previous performed research works and do not focus on the mechanisms by which EGCG can interfere with specific biochemical pathways. More information about the mechanisms should be added in the paper.
2. Section 2.1 (Structure of EGCG) contains some information, such as molecular type and molecular weight, which should be removed, since they do not keep up with the nature of a scientific review.
3. Text under figures 1-4 should be removed, as it only summarizes abovementioned information.
4. Figure 5 is cited in section 3.1, but we see it in the end of section 3.5. It should be removed to the right position.
Comments on the Quality of English LanguageEnglish ediiting required.
Author Response
Dear reviewer,
Thank you for your careful and kind feedback on our manuscript. We have revised the manuscript based on your comments and have carefully proofread the manuscript to minimize typographical, grammatical, and referencing errors. The changes made in response to the reviewers' comments are detailed below.
Comment from Reviewer:
1. The authors review the role of EGCG in Atherosclerosis. In most parts of their paper they only present results from previous performed research works and do not focus on the mechanisms by which EGCG can interfere with specific biochemical pathways. More information about the mechanisms should be added in the paper.
Response: Thank you for your question. In response to your question, we have made the following answers and changes:
Lines 133-139: “Ge Hu et al. [25] showed that EGCG inhibited cholesterol synthesis by down-regulating the activities of mevalonate kinase (MVK), mevalonate 5-pyrophosphate decarboxylase (MDD), and farnesyl pyrophosphate synthase (FPPS) in the mevalonate pathway. In addition, EGCG can reduce TG by directly inhibiting fatty acid synthase (FAS) activity or by down-regulating peroxisome proliferator-activated receptor γ (PPARγ) and FAS expression levels through PI3K-AKT signaling [26].”
Lines 153-160:“Goto et al. [31] found that 25 μMol·L-1 of EGCG up-regulated the expression of the LDL receptor and lowered the apoB100 level to improve the metabolism and thus the metabolism of cholesterol. In vitro biochemical assays, catechins, especially EGCG, inhibited the oxidation of plasma LDL. Choi et al. [32] found that EGCG (25 μMol·L-1) inhibited ox-LDL-induced ROS production, thereby protecting human vascular endothelial cells. Li et al. [33] demonstrated that EGCG (10-100 μMol·L-1) decreased AngII-induced NADPH oxidase expression, lowered ROS generation, and repressed AngII-induced activation of NF-κB and protein factor-1 (AP-1).”
Lines 199-202: “It has been shown that GTPs (containing EGC, EGCG, etc.) lower the abundance of Mycobacterium anisopliae and Clostridium spp. and add the abundance of Mycobacterium thick, decrease the expression of the inflammatory factors IL-6, TNF-α, and IL-1β, and restrain the induction level of the inflammatory signaling pathway toll-like receptor 4 (TLR4) [40].”
Lines 251-255: “EGCG protects vascular endothelial cells from oxidative stress-induced injury by targeting the autophagy-dependent PI3K-AKT-mTOR pathway [47]. Xuan et al. [51] found that EGCG (10 mg·L-1) could protect ECs by activating the PI3K/Akt signaling pathway, increasing the NO level of vascular ECs, and decreasing the level of caspase-3 in the cells.”
In addition, Figure 4 reveals the effects of EGCG on NF-κB, MAPK, and Notch signaling pathways. ERK1/2, JNK, and P38 kinases play key roles in the MAPK pathway, and EGCG inhibited the activation of the MAPK signaling pathway. EGCG induced a significant up-regulation of Tollip in the TLR4 signaling pathway and effectively inhibited the activation of the NF-κB signaling pathway. EGCG induced significant up-regulation of Tollip in the TLR4 signaling pathway and effectively inhibited the activation of NF-κB signaling pathway. The notch signaling pathway has a cross-talk with PI3K/Akt and NF-κB signaling pathways, and EGCG can inhibit the Notch signaling pathway by directly binding to the Notch receptor.
Lines 387-390:”Kim et al. [75] found that epigallocatechin gallate resulted in p21/WAF1-mediated cell cycle G(1) phase arrest and inhibited TNF-α-induced matrix metalloproteinase-9 expression in vascular smooth muscle cells.”
Lines 423-427: “EGCG has been reported to inhibit platelet activity by several mechanisms, including inhibition of collagen-mediated phospholipase (PL) CGA m2, blockade of protein tyrosine phosphorylation, and enhancement of Ca2(+)-ATPase activity, thereby reducing platelet aggregation and attenuating thrombosis [85].”
2. Section 2.1 (Structure of EGCG) contains some information, such as molecular type and molecular weight, which should be removed, since they do not keep up with the nature of a scientific review.
Response: Thank you for your advice on this issue, in order to maintain the rigor of the scientific review. We have removed the information on the molecular type, molecular formula and molecular weight of EGCG from the structure of EGCG in section 2.1 and retained its molecular structure and characteristics.
3. Text under figures 1-4 should be removed, as it only summarizes abovementioned information.
Response: Thank you for your suggestions on this issue, we have removed the text below figures 1-4.
4.Figure 5 is cited in section 3.1, but we see it in the end of section 3.5. It should be removed to the right position.
Response: Thank you for your advice on this issue. The placement of Figure 5 does not make much sense, as it is a summary of the overall improvement of EGCG, of which 3.5 is only one chapter. To address this issue we have added a chapter (Chapter 4) to summarize the anti-atherosclerotic mechanisms of EGCG, and we have also addressed the shortcomings of EGCG and added some references for better research.
Reviewer 3 Report
Comments and Suggestions for Authors
This studies comprehensively review the investigated effects and related mechanisms of EGCG in the prevention and development of atherosclerosis (AS).
The study is well organized, provided with effective explanatory diagrams (Figure 5 is especially useful to summarize), and well written. This review can contribute as a reference for further investigations on EGCG as an interesting molecule for preventing and treating various illnesses and especially AS.
Few suggestions below to improve the manuscript and make it even more attractive.
- Line 68. Abbreviation "GA" (Gallic Acid) is not defined.
- Lines 78-80: to be removed, it's a repetition.
- Line 100: "precaution" - change to "prevention" (here and below).
- Line 111. Intoducing Figure 5 now could hinder the streamline of the text. I suggest omitting.
- Lines 125-126. It is necessary to explain how the antioxidant effect would ameloriate dyslipidemia. It is explained - and complex mechanisms are involved - from line 144 onwards, thus i suggest removing this statement here, where it is unsupported.
- Line 454. Most of the text from here onwards should be moved to a separate section/subsection before the Conclusions.
- Line 469. "Encapsulation" could be added as a potential effective method to improve the resistance and bioavailability. With keywords ("egcg" and "encapsulation"), many thousands of articles can be found. Definitely, EGCG encapsulation should be shortly discussed (before the Conclusions!).
- More general: a list of plant sources of EGCG, beyond tea leaves, with respective concentration (% w/w dry basis) would be useful for future reference.
Comments on the Quality of English LanguageEnglish language is fine.
Author Response
Dear reviewer,
Thank you for your careful and kind feedback on our manuscript. We have revised the manuscript based on your comments and have carefully proofread the manuscript to minimize typographical, grammatical, and referencing errors. The changes made in response to the reviewers' comments are detailed below.
Comment from Reviewer:
This studies comprehensively review the investigated effects and related mechanisms of EGCG in the prevention and development of atherosclerosis (AS).
The study is well organized, provided with effective explanatory diagrams (Figure 5 is especially useful to summarize), and well written. This review can contribute as a reference for further investigations on EGCG as an interesting molecule for preventing and treating various illnesses and especially AS.
Few suggestions below to improve the manuscript and make it even more attractive.
- Line 68. Abbreviation "GA" (Gallic Acid) is not defined.
Response: Firstly, we would like to thank you for your endorsement of this review and for your suggestions. We have removed the abbreviated word EGCG from the title. We have provided the full name of the GA.Please see line 72.
- Lines 78-80: to be removed, it's a repetition.
Response: We have removed the text below that figure.
- Line 100: "precaution" - change to "prevention" (here and below).
Response: We have changed "precaution" to "prevention" in two places throughout the text.
- Line 111. Intoducing Figure 5 now could hinder the streamline of the text. I suggest omitting.
Response: Thanks to your suggestion, this paragraph really should not be here, as Figure 5 is a summary of EGCG's improvement of AS, and 3.1 and 3.5 are just one of the chapters. Therefore, we have removed this paragraph from the row and added a section (Chapter 4) to summarize the potential mechanisms of EGCG for preventing AS as well as the shortcomings of EGCG itself, in order to better explain and organise the paper.
- Lines 125-126. It is necessary to explain how the antioxidant effect would ameloriate dyslipidemia. It is explained - and complex mechanisms are involved - from line 144 onwards, thus i suggest removing this statement here, where it is unsupported.
Response: Thank you for your suggestion, we have removed this sentence from the paragraph.
- Line 454. Most of the text from here onwards should be moved to a separate section/subsection before the Conclusions.
Response: Thank you for the suggestion to add a fourth chapter on the subject as I said above, moving the shortcomings of EGCG itself that existed in the original conclusion section to chapter 4.
- Line 469. "Encapsulation" could be added as a potential effective method to improve the resistance and bioavailability. With keywords ("egcg" and "encapsulation"), many thousands of articles can be found. Definitely, EGCG encapsulation should be shortly discussed (before the Conclusions!).
Response: Thank you for your suggestions regarding the mode of EGCG administration, we have added relevant content and references at the end of Chapter 4, briefly analyzing encapsulation as a potential method that can be used to improve the bioavailability of EGCG. Please see lines 455-462: "Studies have shown that nanocarriers have proven to be excellent materials for encapsulating phenolic compounds and enhancing their bioavailability, including lipid nanoparticles, protein nanoparticles, micelles, emulsions, and metal nanoparticles [93]. By loading phenolic compounds into nanoparticles, not only can their bioavailability be improved, but also targeted release and protection of the active substance can be achieved [94]. Despite the fact that nanoparticles are almost a perfect carrier, there is still a need to consider and minimize their toxicity and side effects. "
- More general: a list of plant sources of EGCG, beyond tea leaves, with respective concentration (% w/w dry basis) would be useful for future reference.
Response: Thank you for your suggestion. In response to your question, we have searched for relevant information and listed the food sources and proportion of EGCG. Since we have only mentioned the proportion of EGCG in catechins and not with tea, we have also added tea (green, black, oolong) as shown in Table 1.
Reviewer 4 Report
Comments and Suggestions for Authors
Review of nutrients-3056732
In the review article, the effect and role of main catechin epigallocatechin gallate (EGCG) in the prevention of atherosclerosis (AS), resisting atherosclerosis and reducing the morbidity and mortality of AS. The anti-AS effects of EGCG on regulation of lipid metabolism, regulation of intestinal flora disorders, modulation of intestinal flora disorders, improvement of vascular endothelial cells functions, inhibition of inflammatory factors expression, regulation of inflammatory signaling pathways, inhibition of matrix metalloproteinase (MMP) expression and inhibition of platelet aggregation are discussed and reviewed. Its role in prevention against lipid metabolism disorders, endothelial damage, inflammation, unstable plaques and the protection of cardiovascular and cerebral blood vessels related to AS, is highly valuated.
To the article, I have next comments and recommendations:
· L. 73: o-dihydroxy catechol: prefix o- should be in Italics.
· L. 96: instead, methyl mercaptan better: methanethiol.
· L. 117: Low-density lipoprotein…
· Unit μMol·L-1 or μmol·L-1: throughout the article should be carefully unified, I prefer μMol·L-1.
· L. 204-205: Mycobacterium anisopliae and Clostridium spp.; Mycobacterium – Latin names should be in Italics.
· L. 284: Cells and elevated serum levels…
· L. 317: … c-Jun N-terminal protein kinase (JNK)…
· In citations in the text throughout the article: e.g., unify Xie et al [65] without a dot or Xie et al. [65] with a dot - this may be preferred.
· L. 367: The Notch signaling pathway…
· L. 419: Lill et al. [80]…
· Reference [9]: use abbreviation of journal´s title: J Cereal Sci.
Author Response
Dear reviewer,
Thank you for your careful and kind feedback on our manuscript. We have revised the manuscript based on your comments and have carefully proofread the manuscript to minimize typographical, grammatical, and referencing errors. The changes made in response to the reviewers' comments are detailed below.
Comment from Reviewer:
In the review article, the effect and role of main catechin epigallocatechin gallate (EGCG) in the prevention of atherosclerosis (AS), resisting atherosclerosis and reducing the morbidity and mortality of AS. The anti-AS effects of EGCG on regulation of lipid metabolism, regulation of intestinal flora disorders, modulation of intestinal flora disorders, improvement of vascular endothelial cells functions, inhibition of inflammatory factors expression, regulation of inflammatory signaling pathways, inhibition of matrix metalloproteinase (MMP) expression and inhibition of platelet aggregation are discussed and reviewed. Its role in prevention against lipid metabolism disorders, endothelial damage, inflammation, unstable plaques and the protection of cardiovascular and cerebral blood vessels related to AS, is highly valuated.
To the article, I have next comments and recommendations:
- L. 73: o-dihydroxy catechol: prefix o- should be in Italics.
Response: Firstly, we would like to thank you for your endorsement of this review and for your suggestions. We have changed the line's prefix o to italics. Please see line 77.
- L. 96: instead, methyl mercaptan better: methanethiol.
Response: We have changed "methyl mercaptan" to "methanethiol" in line 98.
- L. 117: Low-density lipoprotein…
Response: We have capitalized the L in line 115.
- Unit μMol·L-1or μmol·L-1: throughout the article should be carefully unified, I prefer μMol·L-1.
Response: We have double-checked the entire text and standardized the format of all μMol·L-1 units in the text.
- L. 204-205: Mycobacterium anisopliaeand Clostridium spp.; Mycobacterium – Latin names should be in Italics.
Response: We have italicized the Mycobacterium anisopliae and Clostridium spp. names in lines 199-200.
- L. 284: Cells and elevated serum levels…
Response: We have capitalized the letter C in line 282.
- L. 317: … c-Jun N-terminal protein kinase (JNK)…
Response: We have corrected the term to kinase. Please see line 315.
- In citations in the text throughout the article: e.g., unify Xie et al [65] without a dot or Xie et al. [65] with a dot - this may be preferred.
Response: We have checked the literature cited in full, and we have uniformly replaced the el with a dot at the end, with el.
- L. 367: The Notch signaling pathway…
Response: In response to comments from other reviewers, we have deleted the description of the paragraph, and in addition we have checked the capitalization of Notch letters throughout the text.
- L. 419: Lill et al. [80]…
Response: We have changed to Lill et al. Please see line 411.
- Reference [9]: use abbreviation of journal´s title: J Cereal Sci.
Response: We have changed the journal name in reference [10] to the abbreviated J Cereal Sci. Please see line 516.
Round 2
Reviewer 2 Report
Comments and Suggestions for Authors
The authors have made all chnges requested, therefore their work can now be published.